# The Role of Psychological Factors and Vaccine Conspiracy Beliefs in Influenza Vaccine Hesitancy and Uptake among Jordanian Healthcare Workers during the COVID-19 Pandemic

**DOI:** 10.3390/vaccines10081355

**Published:** 2022-08-19

**Authors:** Malik Sallam, Ramy Mohamed Ghazy, Khaled Al-Salahat, Kholoud Al-Mahzoum, Nadin Mohammad AlHadidi, Huda Eid, Nariman Kareem, Eyad Al-Ajlouni, Rawan Batarseh, Nidaa A. Ababneh, Mohammed Sallam, Mariam Alsanafi, Srikanth Umakanthan, Ala’a B. Al-Tammemi, Faris G. Bakri, Harapan Harapan, Azmi Mahafzah, Salah T. Al Awaidy

**Affiliations:** 1Department of Pathology, Microbiology and Forensic Medicine, School of Medicine, The University of Jordan, Amman 11942, Jordan; 2Department of Clinical Laboratories and Forensic Medicine, Jordan University Hospital, Amman 11942, Jordan; 3Department of Translational Medicine, Faculty of Medicine, Lund University, 22184 Malmö, Sweden; 4Tropical Health Department, High Institute of Public Health, Alexandria University, Alexandria 21561, Egypt; 5School of Dentistry, The University of Jordan, Amman 11942, Jordan; 6Cell Therapy Center (CTC), The University of Jordan, Amman 11942, Jordan; 7Department of Pharmacy, Mediclinic Welcare Hospital, Mediclinic Middle East, Dubai P.O. Box 31500, United Arab Emirates; 8Department of Pharmacy Practice, Faculty of Pharmacy, Kuwait University, Kuwait City 25210, Kuwait; 9Department of Pharmaceutical Sciences, Public Authority for Applied Education and Training, College of Health Sciences, Safat 13092, Kuwait; 10Department of Para-Clinical Sciences, Faculty of Medical Sciences, The University of the West Indies, St. Augustine BB11000, Trinidad and Tobago; 11Migration Health Division, International Organization for Migration (IOM), The UN Migration Agency, Amman 11953, Jordan; 12Department of Internal Medicine, School of Medicine, The University of Jordan, Amman 11942, Jordan; 13Infectious Diseases and Vaccine Center, The University of Jordan, Amman 11942, Jordan; 14Medical Research Unit, School of Medicine, Universitas Syiah Kuala, Banda Aceh 23111, Indonesia; 15Tropical Disease Centre, School of Medicine, Universitas Syiah Kuala, Banda Aceh 23111, Indonesia; 16Department of Microbiology, School of Medicine, Universitas Syiah Kuala, Banda Aceh 23111, Indonesia; 17Office of Health Affairs, Ministry of Health, P.O. Box 393, Muscat 100, Oman; 18Middle East, Eurasia and Africa Influenza Stakeholders Network (ME’NA-ISN), Cape Town 7766, South Africa

**Keywords:** health professional, healthcare personnel, medicine practitioner, flu, barrier, vaccine behavior, vaccination intention, vaccine acceptance, seasonal influenza, influenza pandemic

## Abstract

Vaccination to prevent influenza virus infection and to lessen its severity is recommended among healthcare workers (HCWs). Health professionals have a higher risk of exposure to viruses and could transmit the influenza virus to vulnerable patients who are prone to severe disease and mortality. The aim of the current study was to evaluate the levels of influenza vaccine acceptance and uptake as well as its determinants, among Jordanian HCWs over the last influenza season of 2021/2022. This study was based on a self-administered electronic survey that was distributed in March 2022. Psychological determinants of influenza vaccine acceptance and vaccine conspiracy beliefs were assessed using the previously validated 5C scale questionnaire (confidence, complacency, constraints, calculation and collective responsibility) and the vaccine conspiracy beliefs scale. The study sample comprised a total of 1218 HCWs: nurses (*n* = 412, 33.8%), physicians (*n* = 367, 30.1%), medical technicians (*n* = 182, 14.9%), pharmacists (*n* = 161, 13.2%) and dentists (*n* = 87, 7.1%), among others. About two-thirds of the study sample expressed willingness to receive influenza vaccination if provided free of charge (*n* = 807, 66.3%), whereas less than one-third were willing to pay for the vaccine (*n* = 388, 31.9%). The self-reported uptake of the influenza vaccine in the last influenza season was 62.8%. The following factors were significantly associated with higher acceptance of influenza vaccination if provided freely, as opposed to vaccine hesitancy/rejection: male sex; physicians and dentists among HCW categories; higher confidence and collective responsibility; and lower complacency, constraints and calculation. Higher influenza vaccine uptake was significantly correlated with nurses and physicians among HCW categories, older age, a higher monthly income, higher confidence and collective responsibility, lower complacency and constraints and lower embrace of general vaccine conspiracy beliefs. The results of the current study can provide helpful clues to improve influenza vaccine coverage among HCWs in Jordan. Consequently, this can help to protect vulnerable patient groups and reserve valuable resources in healthcare settings. Psychological determinants appeared to be the most significant factors for vaccine acceptance and uptake, whereas the embrace of general vaccine conspiracy beliefs was associated with lower rates of influenza vaccine uptake, which should be considered in educational and interventional measures aiming to promote influenza vaccination.

## 1. Introduction

Seasonal influenza was implicated in significant morbidity and mortality prior to the coronavirus disease 2019 (COVID-19) pandemic [1,2]. Specifically, about 3–8 million cases of severe illness were reported in 2017 worldwide, with 290,000–650,000 deaths attributed to seasonal influenza annually during the period of 1999–2015 [2,3]. The widespread adoption of non-pharmaceutical intervention (NPI) measures for the prevention of COVID-19 affected other respiratory viruses, including influenza, reducing its burden, besides the possible antagonistic role of viral interference to prevent concomitant infection by other viruses [4,5]. On the other hand, the waning population immunity to influenza could result in epidemic surge with severe consequences in the upcoming winter seasons [6,7].

The pandemic potential of influenza virus is well documented, with the last recorded influenza pandemic taking place in 2009 [1,8]. This pandemic potential of influenza is related to the swift evolutionary changes in the viral genome due to lack of proofreading activity in the replicating enzyme, as well as the segmented nature of the genome [9,10]. The devastating consequences of pandemics were manifested clearly in the 1918–1920 pandemic (Spanish flu), which killed 20–50 million people [11,12].

The risk of increased morbidity and mortality from influenza is observed among specific groups: (1) children and the elderly and (2) individuals with comorbidities (e.g., asthma, heart disease, stroke, diabetes mellitus and chronic kidney disease) [13,14].

Healthcare workers (HCWs) can be viewed as a risk group for an increased likelihood of influenza virus acquisition due to a higher exposure [15,16]. Subsequently, influenza among HCWs can pose risks for patients, colleagues and relatives; therefore, influenza vaccination of HCWs can improve patient safety [17,18,19,20]. Additionally, influenza infection among HCWs can reduce their productivity and increase absenteeism, with subsequent high economic burden [21,22]. On the other hand, a few studies and scientific reviews have pointed to the absence of unambiguous influenza vaccination benefits among HCWs rationalizing the controversy surrounding the issue of mandatory influenza vaccination in this key target group [23,24].

Vaccination is considered the primary preventive measure to prevent influenza or alleviate its adverse health and economic impact [25]. Based on the latest World Health Organization (WHO) recommendations, the following groups should be prioritized for annual influenza vaccination before the beginning of the influenza season: (1) HCWs, (2) pregnant women, (3) individuals with certain chronic diseases for more than six months, (4) elderly people over the age of 65 years, (5) children aged six months to five years and (6) residents of institutions for older persons and the disabled [26].

Several barriers challenge the successful implementation of influenza vaccination programs, particularly in low- and middle-income countries [27,28]. These barriers include a lack of clear data regarding the burden of influenza; competing health priorities, compromising financial support for the vaccination program; and vaccine hesitancy, forming a major obstacle to influenza vaccine uptake [29,30,31].

Vaccine hesitancy as a phenomenon has its roots in the early days of vaccination inception, and it has gained attention during the ongoing COVID-19 pandemic [32,33,34,35]. It is defined as the reluctance to receive vaccines, even if vaccination services are available [36]. Efforts to understand vaccine hesitancy have pointed to the importance of several psychological factors often modeled by the 5C or 7C models, indicating the relevance of confidence, complacency, constraints, calculation, collective responsibility, compliance and conspiracy beliefs [37,38,39,40]. Thus, the inspection of these factors can help to devise well-informed educational programs and interventional measures that might help to increase vaccine uptake [37].

Accordingly, the success of vaccination programs is dependent upon vaccination behavior that is influenced by several psychological antecedents, as follows: (1) confidence, which involves the level of trust in vaccine safety and efficacy and trust in health professionals and official sources that recommend vaccination; (2) complacency, which is determined by the perception of disease risks, with higher levels correlated with decreased willingness to get vaccinated; (3) constraints, which involve psychological and/or physical barriers, as well as cost issues associated with vaccine hesitation; (4) calculation, which involves weighing possible benefits and risks from vaccination; (5) collective responsibility, which entails the desire to protect the vulnerable members of society from the dire consequences of infections; (6) compliance, which involves the readiness to support vaccination initiatives and instructions, including the mandates that punish unvaccinated individuals; and (7) conspiracy beliefs regarding vaccines and the embrace of vaccine misinformation [37,38,39,40,41,42].

Regarding influenza vaccine uptake and hesitation, HCWs are a key target group that is considered among the most studied groups worldwide [43,44,45,46,47]. The necessity of annual accommodation of new vaccine formulations confers a peculiarity of influenza vaccination and raises questions about declining vaccine effectiveness, leading to vaccine hesitancy [48,49,50]. Determinants of hesitation towards influenza vaccination among HCWs include psychological, physical, contextual and sociodemographic barriers, which were comprehensively presented in a systematic review by Schmid et al. [43]. The relevance of this review is related to the thorough presentation of possible barriers to influenza vaccination on micro and macro levels [43]. The review establishes a clear direction for future research that can help in the design of evidence-based intervention measures to address influenza vaccination hesitancy [43].

Psychological barriers to influenza vaccine acceptance and uptake involve the following: (1) utility of vaccination by assessing benefits and risks; (2) risk perception for the disease and the vaccine at the cognitive and affective levels; (3) social benefit; (4) subjective norms associated with the influence of the perceived pressure of others to get vaccinated and the belief that vaccination is an ethical or professional obligation; (5) perceived behavioral control, with an individual evaluation of one’s own capacity to adopt a certain behavior; (6) attitude; (7) past behavior; (8) experience, including a previous history of experiencing influenza and the duration of experience in the healthcare profession; and (9) the level of knowledge about the disease [43].

Physical barriers include lifestyle (smoking, alcohol consumption and level of physical activity), body mass index and history of chronic disease. Contextual barriers include access to vaccination services (geographic and economic); interaction with the healthcare system (e.g., having a source of care); cues of action, involving direct recommendations by experts; and system factors, involving the size of a healthcare facility. Sociodemographic factors include age, sex, race and marital status [43].

In Jordan, and similarly in a majority of countries in the Middle East and North Africa (MENA), the phenomenon of vaccine hesitancy drew special attention during the ongoing COVID-19 pandemic [32,41,51]. The reported rates of COVID-19 vaccine acceptance in the MENA region were among the lowest worldwide, with significant association with high rates of embracing vaccine conspiracy beliefs and circulation of misinformation about the pandemic [32,33,41,52,53]. In addition, low rates of influenza vaccine acceptance were reported among the general public and university students amid the ongoing pandemic in Jordan [41,54].

Thus, the aim of this study was to estimate the rate of influenza vaccine uptake in the last influenza season (2021/2022) among Jordanian HCWs. In addition, we aimed to evaluate influenza vaccine acceptance and uptake and their associated psychological determinants among various HCW categories in the country. Moreover, we sought to investigate the effect of the requirement to pay for influenza vaccination on vaccine acceptance among the study group. Finally, a study objective was to investigate the possible association between influenza vaccine uptake/acceptance and the embrace of general vaccine conspiracy beliefs.

## 2. Materials and Methods

### 2.1. Study Design

This cross-sectional survey was based on data collected from the population of interest, namely Jordanian HCWs at one point in time starting on 17 March 2022 and finishing on 31 March 2022. We used Google Forms as the survey administration platform. Recruitment of participants was based on the chain-referral sampling approach. Sampling began by sharing the survey on social media platforms (Facebook, Twitter and Instagram) and the free messaging service WhatsApp, as well as sending the survey to contacts of the authors in Jordan and asking the participants to share the survey link among their colleagues.

The inclusion criteria, as explicitly mentioned in the introductory section of the survey link, were (1) Jordanian HCWs and (2) age of 22 years or older. The survey was distributed in the Arabic language, with no incentives for participation. Response to all items was mandatory to avoid incomplete data, which may inversely impact the benefit of the employed scales. The minimum sample size was calculated based on an estimate of approximately 100,000 HCWs in Jordan [55,56]. We used the online tool “Epitools—Epidemiological Calculators”, with the following preset parameters: estimated proportion of 0.5, desired precision of estimate (margin of error) of 3% and 95% confidence level (CL) [57,58]. The final calculation was based on the following formula: *n* = (Z2 × P × (1 − P))/e^2^, where Z = value from standard normal distribution corresponding to the desired confidence level (Z = 1.96 for 95% CL), P is the expected true proportion and e is the desired precision (half desired CL width). Based on the previous calculations, we found that 1057 respondents would represent the minimum required sample size in this study.

The study was approved by the Scientific Research Committee at the School of Medicine/University of Jordan (reference number: 1466/2022/67). An electronic informed consent was ensured by the presence of a mandatory item in the introductory part of the survey: “Do you agree to participate in this study?”.

### 2.2. Survey Instrument

The items used in the survey were adopted from previous studies as follows. Survey items assessing knowledge of influenza were adopted from Al Awaidy et al. [46], whereas items assessing the 5C psychological determinants of vaccination were adopted from the previously validated and reliable tool in the Arabic language by Abd ElHafeez et al. [59]. Assessment of the attitude toward the general embrace of vaccine conspiracy beliefs was based on a previously validated instrument, the “vaccine conspiracy beliefs scale (VCBS)”, by Shapiro et al. [60].

The online survey comprised five sections: first, an introductory section with information about the study and its objectives, followed by an item to ensure provision of an e-consent for participation. If the respondent selected “No” for this item, the questionnaire closed immediately.

Second, an eight-item section assessing the demographics of the respondents was presented as follows: (1) sex (male vs. female), (2) age (18–75), (3) nationality (Jordanian vs. non-Jordanian), (4) occupational category (nurse; physician; dentist; pharmacist; medical technician, including professions in laboratory, radiology, rehabilitation and anesthesia; or medical secretaries, administrators and receptionists (MSAR)), (5) residence (the capital, Amman; Irbid; Zarqa; Mafraq; Ajloun; Jerash; Madaba; Balqa; Karak; Tafilah; Ma’an; or Aqaba), (6) educational level (undergraduate vs. postgraduate), (7) monthly household income (<1000 Jordanian dinar (JOD, equal to USD ~ 1410) vs. JOD ≥ 1000) and (8) history of chronic disease (yes vs. no).

Third, an 11-item section on influenza knowledge (6 items) and attitude/practice (five items) toward influenza vaccination was presented as follows: (1) influenza causes mild symptoms only; therefore, it cannot be considered a serious disease (yes vs. no); (2) influenza can cause severe illness or death (yes vs. no); (3) influenza can be transmitted through droplets and aerosols from coughing or sneezing (yes vs. no); (4) influenza can be transmitted via blood and body fluids (yes vs. no); (5) influenza vaccination reduces absenteeism from work (yes vs. no); (6) vaccinating healthcare workers against influenza helps to protect patients from severe illness or death (yes vs. no); (7) in Jordan, vaccination of healthcare workers against influenza should be compulsory (yes vs. no); (8) are you willing to get the influenza vaccine if it were available for free? (yes vs. maybe vs. no); (9) are you willing to pay an amount not exceeding JOD 15 to get the influenza vaccine? (yes vs. maybe vs. no); (10) do you recommend influenza vaccination for patients (yes vs. no vs. not applicable); and (11) have you received an influenza vaccine in the previous year? (yes vs. no). The first six knowledge items were used to calculate a “flu knowledge score”, with one point awarded for each correct answer, yielding a maximum score of six.

Fourth, the questionnaire included a 15-item section on the 5C psychological determinants of influenza vaccination, assessed using a 7-point Likert scale (strongly disagree, disagree, disagree to some extent, neutral, agree to some extent, agree and strongly agree). The items that assessed confidence included: (1) I am absolutely confident that the influenza vaccine is safe; (2) the influenza vaccine is effective; and (3) as for vaccination, I am sure that the public authorities decide what is best for the good of society. The items that assessed complacency included: (1) influenza vaccination is not necessary because influenza is not as common as it used to be; (2) my immune system is very strong, and it will protect me from influenza; and (3) influenza is not serious enough for me to get the vaccine. The items that assessed convenience included: (1) daily pressure prevents me from getting vaccinated; (2) for me, it is inconvenient to get the influenza vaccine; and (3) I feel uncomfortable when visiting a doctor, and this makes me avoid vaccination. The items that assessed calculation included: (1) when I think about getting influenza vaccine, I consider the risks and benefits to make the best possible decision; (2) for vaccination, I think carefully about its benefits for me; and (3) before I get vaccinated, it is very important for me to fully understand everything about vaccination. Finally, the items that assessed collective responsibility included: (1) when everyone gets vaccinated, I do not have to get the vaccine myself; (2) I get the vaccine to protect the community members with the weakest immunity; and (3) vaccination is a collective action to prevent the spread of disease.

The final section comprised seven items that assessed the general embrace of vaccine conspiracy beliefs using a 7-point Likert scale (strongly disagree, disagree, disagree to some extent, neutral, agree to some extent, agree and strongly agree). The items included: (1) vaccine safety data are often fabricated; (2) vaccination of children is harmful, and this fact is hidden from people; (3) pharmaceutical companies cover up the dangers of vaccines; (4) people are deceived about the effectiveness of vaccines; (5) vaccine efficacy data are often fabricated; (6) people are deceived about the safety of vaccines; and (7) the government is trying to cover up the link between vaccines and other diseases, such as autism. Lower VCBS scores were correlated with a lower embrace of general vaccine conspiracy beliefs.

### 2.3. Main Study Measures

The major outcome measures in this study were: (1) willingness to get influenza vaccination if provided freely, with responses dichotomized as: yes, indicating acceptance vs. no/maybe, indicating hesitancy; (2) willingness to get influenza vaccination if payment is needed, with responses dichotomized as: yes, indicating acceptance vs. no/maybe, indicating hesitancy; and (3) influenza vaccine uptake in the last influenza season (yes vs. no). The possible correlated factors (each 5C subscale and VCBS) were dichotomized based on the mean value of each subscale variable as follows: (1) confidence subscale: <8 vs. ≥8; (2) complacency subscale: <11 vs. ≥11; (3) constraints subscale: <9 vs. ≥9; (4) calculation subscale: <18 vs. ≥18; (5) collective responsibility subscale: <7 vs. ≥7; and (6) VCBS: <24 vs. ≥24. The covariates were sex, age (≤38 years vs. >38 years), occupational category (physicians vs. nurses vs. dentists vs. pharmacists vs. medical technicians, excluding the MSAR due to a very limited number of participants in this category), educational level (undergraduate vs. postgraduate) and monthly income level (JOD < 1000 vs. JOD ≥ 1000).

### 2.4. Statistical Analysis

Statistical analysis was performed using IBM SPSS Statistics for Windows, version 22.0. (Armonk, NY, USA: IBM Corp.). Chi-squared test (χ^2^) was used to assess the differences in influenza knowledge and occupational category. To assess the association between the “flu knowledge score” and the categorical variables, we used the Mann–Whitney *U* test (M–W) and the Kruskal–Wallis (K–W) test. Multinomial regression analyses were used to investigate the predictors of influenza vaccine acceptance and uptake. The statistical significance was determined at the 0.050 cut-off.

## 3. Results

### 3.1. General Features of the Study Sample

The total number of surveys that were filled was 1342. We applied a filtration algorithm to remove the responses from those who did not consent to participate, those who did not meet the inclusion criteria and those with apparently careless responses (Appendix A). The final number of HCWs enrolled in this study was 1218, with a predominance of females (60.3%) and a median age of 38 years (interquartile range (IQR): 31–49 years, Table 1).

Nurses and physicians comprised slightly less than two-thirds of the study sample, whereas medical secretaries, administrators and receptionists (MSAR) represented less than 1.0% of the study respondents (Table 1).

### 3.2. Knowledge of Influenza and Attitude toward Its Vaccination among the Study Respondents

The overall level of knowledge regarding influenza is illustrated in (Figure 1). The general level of correct responses was above 75.0% for five out of six knowledge items, whereas only 53.7% (*n* = 654) of the study sample correctly responded to the item “Influenza causes only mild symptoms; therefore, it cannot be considered a serious disease”. About half of the study sample responded that vaccination of HCWs against influenza should be mandatory in Jordan (*n* = 585, 48.0%).

Stratified per occupational category, physicians appeared to have a higher level of knowledge compared to other categories for the majority of influenza knowledge items (Table 2).

With respect to the “flu knowledge score”, the following categories had a significantly higher score: participants older than 38 years (mean score: 4.87 vs. 4.74, *p* = 0.017, M–W), physicians (mean score: 5.06 vs. 4.82 among dentists vs. 4.73 among nurse vs. 4.70 among pharmacists vs. 4.56 among MSAR vs. 4.55 among medical technicians, *p* < 0.001, K–W), residents in the capital, Amman (mean score: 4.87 vs. 4.68, *p* = 0.001, M–W), participants with postgraduate degrees (mean score: 4.90 vs. 4.74, *p* = 0.003, M–W) and participants with monthly income JOD ≥1000 (mean score: 4.90 vs. 4.70, *p* < 0.001, M–W).

A higher flu knowledge score was observed among participants who were willing to get influenza vaccination if provided freely compared to those who were hesitant and resistant (mean score: 4.95 vs. 4.67 vs. 4.25, *p* < 0.001, K–W). Additionally, a higher score was observed among participants who were willing to pay for the vaccine compared to those who were hesitant or resistant (mean score: 4.99 vs. 4.90 vs. 4.58, *p* < 0.001, K–W). The participants who reported uptake of influenza vaccine in the last influenza season also had a higher mean flu knowledge score (mean score: 4.85 vs. 4.73, *p* = 0.037, M–W).

When applicable, the participants who reported recommending influenza vaccination to patients had a higher mean flu knowledge score (4.93 vs. 4.17, *p* < 0.001, M–W). The same pattern was observed by classifying the study sample based on occupational category, with the exception of pharmacists (mean score among physicians: 5.09 vs. 4.67, *p* = 0.034, M–W; mean score among nurses: 4.87 vs. 4.15, *p* < 0.001, M–W; mean score among dentists: 5.04 vs. 3.67, *p* = 0.001, M–W; mean score among medical technicians: 4.77 vs. 3.90, *p* < 0.001, M–W; mean score among pharmacists: 4.71 vs. 4.35, *p* = 0.223, M–W).

### 3.3. About Two-Thirds of the Study Sample Reported Influenza Vaccine Uptake in the Last Season

The self-reported influenza vaccine uptake among the study respondents was 62.8% (*n* = 765). The highest percentage of influenza vaccine uptake was reported among nurses (70.9%), followed by physicians (67.8%), dentists (59.8%), pharmacists (52.8%) and medical technicians (47.3%), whereas the lowest rate was observed among medical secretaries, administrators and receptionists (11.1%, *p* < 0.001, χ^2^ = 51.867, Table 3).

A higher percentage of influenza vaccine uptake was reported among male respondents (68.3% vs. 59.2% in female respondents, *p* = 0.001, χ^2^ = 10.422), participants with a postgraduate degree (68.3% vs. 59.4% among participants with an undergraduate degree, *p* = 0.002, χ^2^ = 9.745) and those with a history of chronic disease (74.5% vs. 60.1%, *p* < 0.001, χ^2^ = 16.565). An older mean age was observed among the participants who reported influenza vaccine uptake compared to those who did not get the vaccine in the last season (mean age 42.6 vs. 38.4 years, *p* < 0.001, M–W). Participants with a higher monthly income had a higher percentage of influenza vaccine uptake; however, this difference lacked statistical significance (64.8% vs. 60.7%, *p* = 0.136, χ^2^ = 2.222). Additionally, participants residing in the capital, Amman, had a higher rate of influenza vaccine uptake compared to those residing outside the capital; nevertheless, this difference lacked statistical significance (63.4% vs. 61.7%, *p* = 0.539, χ^2^ = 0.378).

### 3.4. Physicians Had the Highest Rate of Vaccine Acceptance, and Nurses Had the Lowest Rate of Willingness to Pay for the Vaccine

Physicians were found to have the highest rate of influenza vaccine acceptance if provided freely, as opposed to vaccine hesitancy/rejection (81.7%), followed by dentists (71.3%), nurses (64.6%) and pharmacists (55.9%), whereas the lowest rate was found among medical technologists (48.9%, *p* < 0.001, χ^2^ test, Table 3).

A higher percentage of influenza vaccine acceptance, if provided freely, was reported among male respondents (74.3% vs. 59.2% in females, *p* < 0.001, χ^2^ = 23.320), participants with a postgraduate degree (71.1% vs. 63.3% among participants with an undergraduate degree, *p* = 0.005, χ^2^ = 7.933), participants with higher monthly income (69.7% vs. 62.5%, *p* = 0.008, χ^2^ = 7.059), those residing in the capital, Amman (68.5% vs. 62.4%, *p* = 0.030, χ^2^ = 4.698), and participants with a history of chronic disease (75.8% vs. 64.0%, *p* = 0.001, χ^2^ = 11.511). An older mean age was observed among participants who accepted influenza vaccination if provided freely compared to those who were hesitant or resistant to vaccination (mean age 41.8 vs. 39.5 years, *p* = 0.014, M–W). The aforementioned comparisons are illustrated in (Table 3), with three possible outcomes (acceptance vs. hesitancy vs. rejection).

With respect to influenza vaccine acceptance if payment is required for the vaccine, physicians were found to have the highest rate of willingness to pay for the vaccine (51.2%), followed by dentists (49.4%), pharmacists (24.8%) and medical technologists (21.4%), whereas the lowest rate was found among nurses (18.9%, *p* < 0.001, χ^2^ = 124.475, Table 3). None of the MSAR participants (*n* = 9) showed a willingness to pay for influenza vaccination.

A higher percentage of willingness to pay for influenza vaccination was reported among males (41.0% vs. 25.9% among females, *p* < 0.001, χ^2^ = 30.791), participants with a postgraduate degree (44.6% vs. 24.0% among participants with an undergraduate degree, *p* < 0.001, χ^2^ = 56.188), participants with higher monthly income (43.0% vs. 20.0%, *p* < 0.001, χ^2^ = 74.091), those residing in the capital, Amman (37.6% vs. 21.8%, *p* < 0.001, χ^2^ test), and participants with a history of chronic disease (46.3% vs. 28.5%, *p* < 0.001, χ^2^ = 27.476). An older mean age was found among the participants who showed willingness to pay for influenza vaccination compared to those who were hesitant or resistant to vaccination (mean age 46.1 vs. 38.7 years, *p* < 0.001, M–W, Table 3).

### 3.5. Influenza Vaccine Acceptance Drops Significantly if Payment Is Required

A significant drop in influenza vaccine acceptance was noticed if payment were required across all possible variables tested, as illustrated in Figure 2. The same pattern was noticed across occupational categories as follows: among nurses, the influenza vaccine acceptance, if provided freely, was 64.6% compared to 18.9% if payment were required (*p* < 0.001, χ^2^ = 176.378); among physicians, the acceptance of free influenza vaccination was 81.7% compared to 51.2% if payment were required (*p* < 0.001, χ^2^ = 76.697); among pharmacists, the acceptance of free influenza vaccination was 55.9%, as opposed to 24.8% if payment were needed (*p* < 0.001, χ^2^ = 32.252); for medical technicians, a drop was noticed from 48.9% for free influenza vaccination to 21.4% if payment were needed (*p* < 0.001, χ^2^ = 30.124); and for dentists, the same drop was observed (71.3% acceptance of free influenza vaccination vs. 49.4% if payment were required, *p* = 0.003, χ^2^ = 8.670).

### 3.6. The 5C Psychological Determinanats Were Associated with Influenza Vaccine Acceptance if Provided Freely

Multinomial regression analysis showed that influenza vaccine acceptance if the vaccine is provided freely was significantly associated with all the 5C psychological determinants (Figure 3). However, the VCBS was not correlated with influenza vaccine acceptance if provided free of charge (*p* = 0.756). Confidence and collective responsibility showed the highest correlation with influenza vaccine acceptance if provided freely (odds ratio (OR): 4.5, 95% confidence interval (CI): 3.2–6.3 for confidence and OR: 4.0, 95% CI: 2.8–5.9, Figure 3), with vaccine hesitancy/rejection as the reference category.

Occupational category (*p* < 0.001, with higher acceptance among physicians and medical technicians as the reference category) and sex (*p* = 0.030, with higher acceptance among male respondents) were the only covariates that were associated with influenza vaccine acceptance if provided freely, whereas age (*p* = 0.507), education (*p* = 0.752) and monthly income (*p* = 0.295) did not show such a correlation.

### 3.7. Confidence, Complacency, Constraints and Collective Responsibility Were Correlated with Willingness to Pay for Influenza Vaccination

Multinomial regression analysis for influenza vaccine acceptance if payment is needed showed that four of the 5C psychological determinants of vaccination were correlated with willingness to pay for influenza vaccination, whereas calculation was the only factor that did not show such an association (*p* = 0.717, Figure 3). Confidence and complacency showed the highest correlation with willingness to pay for influenza vaccination (OR: 3.4, 95% CI: 2.4–4.7 for confidence and OR: 2.5, 95% CI: 1.8–3.5 for complacency). However, the VCBS was not correlated with willingness to pay for the vaccine (*p* = 0.530).

Age (*p* = 0.002, with a higher rate among participants older than 38 years) and monthly income (*p* = 0.005, with a higher rate among participants with monthly income JOD ≥1000) were the only covariates that were associated with willingness to pay for influenza vaccination, whereas sex (*p* = 0.170), education (*p* = 0.143) and occupational category (*p* = 0.523) did not show such a correlation.

### 3.8. Confidence, Complacency, Constraints, Collective Responsibility and Embrace of Vaccine Conspiracy Beliefs Were Correlated with Influenza Vaccine Uptake

Multinomial regression analysis showed that the actual influenza vaccine uptake was significantly correlated with four of the 5C psychological determinants of vaccination (Figure 3).

In particular, constraints was found to be the major 5C factor associated with actual influenza vaccine uptake (OR: 2.5, 95% CI: 1.8–3.3), followed by complacency (OR: 2.1, 95% CI: 1.5–2.8), confidence (OR: 1.8, 95% CI: 1.3–2.4) and collective responsibility (OR: 1.5, 95% CI: 1.1–2.0), whereas calculation was the only factor that did not show such an association (*p* = 0.649, Figure 3).

Additionally, the embrace of general vaccine conspiracy beliefs, as evidenced by higher VCBS scores, was correlated with lower influenza vaccine uptake (OR: 0.71, 95% CI: 0.52–0.97, *p* = 0.034).

The participants who reported getting the influenza vaccination in the last influenza season (2021), had a lower mean VCBS compared to those who did not receive the vaccine (mean VCBS: 22.5 vs. 25.7, *p* < 0.001, M–W).

Occupational category (*p* < 0.001, with higher uptake among nurses and physicians, with medical technicians as the reference category), age (*p* = 0.032, with higher uptake among participants older than 38 years) and monthly income (*p* = 0.040, with higher uptake among participants with monthly income JOD ≥ 1000) were the only covariates that were associated with uptake of influenza vaccination, whereas sex (*p* = 0.316) and education (*p* = 0.197) did not show such a correlation.

## 4. Discussion

The major findings of this study can be summarized in the following five points. First, a majority of Jordanian HCWs that were sampled in this study would accept influenza vaccination if provided freely; nevertheless, one-third of the study respondents were hesitant/resistant to influenza vaccination. Second, influenza vaccine acceptance would drop significantly if HCWs were required to pay for vaccination across the tested variables. Third, influenza vaccine uptake during the last influenza season (2021/2022) was 63%, which is slightly higher or close to the rates observed in other countries in the Middle East [61,62]. Nevertheless, the current uptake level can be improved to protect HCWs and, subsequently, their patients from the negative impact of influenza. Fourth, the psychological determinants of vaccine acceptance, namely confidence, complacency, convenience, calculation and collective responsibility, were significantly correlated with influenza vaccine acceptance among Jordanian HCWs. Finally, general vaccine conspiracy beliefs did not appear to be decisive with respect to the decision to accept influenza vaccination among Jordanian HCWs; however, this factor was significantly correlated with the actual uptake of influenza vaccine among the study group.

Compared to a recent study that was conducted among HCWs in Jordan in 2016, a higher rate of influenza vaccine coverage was noted in this study (63% vs. 53%) [63]. The influenza vaccine uptake rate was merely 18% among the general public in Jordan as reported in an earlier study dating back to 2012 [64]. One possible explanation for this increasing trend in influenza vaccine uptake is increased awareness regarding the importance of vaccination against infectious respiratory diseases during the COVID-19 pandemic, especially for the HCWs, in view of their frontline position in the fight against the disease [65,66]. A similar pattern was reported in a recent multicenter study conducted in Italy, where the proportion of influenza-vaccinated HCWs increased from 4% in 2013–2014 to 54% in the 2020–2021 influenza season [67]. An increasing trend in influenza vaccine coverage among HCWs was also reported in several studies from Italy and Greece [68,69,70,71]. Another possible reason for the finding of increased influenza vaccine uptake in Jordan might be the intensified governmental campaigns to raise the awareness of the importance of influenza vaccination and the quest to reduce barriers that could hinder vaccine uptake, including a reduction in influenza vaccine cost in the country [72].

Globally, an extensive variability in influenza vaccine coverage was reported across different regions [73,74,75,76,77]. This could be related to different policies adopted regarding influenza vaccination, including those related to compulsory vaccination, in addition to challenges associated with the procurement, storage and distribution of vaccines, particularly in low- to middle-income countries [28,78].

In this study, the strongest predictors of influenza vaccine acceptance were high levels of confidence and collective responsibility, followed by low levels of complacency, constraints and calculation. This result is consistent with the findings of several recent studies that highlighted the significant role of psychological factors in attitude toward vaccination both among the general public and among HCWs [43,79,80]. For example, confidence manifested as a trust in government was found to be a significant factor in willingness to accept a vaccine that is approved as safe and effective in a recent study conducted in Hungary, Israel and Japan [81]. The role of vaccine confidence in actual uptake of influenza vaccination was also evident in a recent study among HCWs in Italy [82]. In addition, high levels of collective responsibility and confidence, as well as low levels of constraints and calculation, were found to be significantly correlated with COVID-19 vaccine acceptance in a recent study among HCWs in Kuwait [83].

Complacency was also found to be a significant factor in influenza vaccine acceptance and uptake. Consistent with this result, a recent study conducted among nurses in Hong Kong showed that lower levels of complacency were linked with higher uptake of influenza vaccination [84]. The impact of complacency in this study could be related to the finding of slightly less than half of the study group stating that influenza causes mild symptoms only; therefore, it cannot be considered as a serious disease. Thus, improving the level of disease knowledge can increase influenza vaccine acceptance and enhance its uptake among HCWs. Tackling the lack of knowledge regarding influenza and its vaccine is recommended, as this finding has often been cited as an important determinant of influenza vaccine acceptance and uptake in recent and past studies [85,86].

The results of the present study demonstrate that higher levels of knowledge about influenza are associated with increased willingness to accept influenza vaccination if provided freely, increased willingness to pay for the vaccine and a higher percentage of actual uptake of the vaccine. Previously, this correlation was evident in a large multicenter study from France that was conducted among HCWs [87]. In the aforementioned study, Kadi et al. suggested that the low influenza vaccine coverage reported in their study was associated with lack of knowledge regarding influenza and its vaccine [87]. A similar pattern was reported in an earlier study among HCWs in the U.S. [88]. Therefore, strategies to improve the level of influenza vaccine acceptance and uptake can benefit from focused educational campaigns among HCWs to improve the level of knowledge about the disease and to highlight the positive impact of vaccination on patient care [89]. Importantly, the differences in the level of knowledge observed among HCW categories should be taken into account in such educational campaigns. Considering that lower levels of influenza knowledge were found among nurses, pharmacists and medical technicians compared to physicians and dentists, the former groups can benefit from more didactic instructional educational sessions [90].

Regarding self-reported influenza vaccine uptake in this study, lower levels of constraints appeared to be the most significant psychological factor among the study respondents. Easy access to vaccination services, including on-site vaccination offered at convenient times, can be helpful in overcoming constraints as a barrier to vaccine acceptance and uptake [91,92]. Provision of free vaccination, providing incentives and frequent reminders (e.g., by short message service) are among the recommended measures to improve vaccine uptake [89,93,94]. However, the motivational role of financial incentives to increase the influenza vaccination coverage was not evident in a recent study among the elderly in Hungary [95]. Thus, future studies are recommended to elucidate the role of monetary incentives in actual influenza vaccine uptake.

The analysis of sociodemographic characteristics in relation to influenza vaccine acceptance and uptake in this study showed that male respondents had higher vaccine acceptance rates. Similar result were reported in various studies, which could be partly related to a higher level of fear of side effects following vaccination among females, particularly in relation to pregnancy [43,96]. Older age was also associated with higher influenza vaccine uptake but not with vaccine acceptance. Efforts to understand the reasons behind such differences are rarely undertaken, which should be addressed in future research [43]. Other sociodemographic variables that were correlated with higher acceptance of influenza vaccination and uptake included higher monthly income and higher educational level, as well as a history of chronic disease. This result is consistent with the results of a majority of similar studies and can be attributed to increased knowledge and affordability, as well as lower levels of complacency among those with a history of chronic disease [43,97,98,99].

The differences observed in the rates of influenza vaccine acceptance and uptake among HCW categories warrant implementation of slightly different strategies for vaccine promotion depending on the group. These strategies are recommended to be tailored according to occupational category [89,100]. In this study, physicians had the highest rates of vaccine acceptance compared to nurses. A similar pattern was reported in two different studies from Saudi Arabia and Italy [101,102,103]. Taken together, the differences in influenza vaccine acceptance based on HCW category should be taken into account in efforts aimed at increasing vaccine uptake [100].

An interesting result of this study is the correlation between self-reported influenza vaccine uptake and lower levels of embrace of general vaccine conspiracy beliefs. This correlation has not been investigated previously to the best of our knowledge. Such a result highlights the importance of incorporating conspiracy as a determinant of vaccine acceptance/uptake, which was recently done in the 7C model for the investigation of vaccine readiness [37]. A similar correlation was observed in our previous research in the context of COVID-19 vaccine hesitancy [41,54,83].

The frequent emergence of infectious diseases, accompanied by the wide prevalence of adopting conspiratorial narratives, has become a notable observation (e.g., during the COVID-19 pandemic and the ongoing multicountry monkeypox outbreak) [104,105,106,107]. The negative impact of conspiracy beliefs related to the COVID-19 pandemic was presented comprehensively in a recent systematic review by Ripp and Röer [104]. This includes the reduced willingness to get vaccinated, in addition to the negative association with preventive behavior (e.g., adhering to social distancing guidelines) [104,108]. In this study, the negative impact of endorsing vaccine conspiracy beliefs was manifested in the association with lower rates of influenza vaccine uptake in the study sample, highlighting the need to consider the aspects of conspiracies in the implementation of effective intervention measures with the aim of promoting vaccination [104,109].

The present study clearly shows the positive impact of providing vaccination for free as opposed to the acceptance of influenza vaccination if payment is needed. In this study, the item that assessed influenza vaccine acceptance if payment were needed was based on the current market price of the vaccine in Jordan (about JOD 15, equal to USD ~21). However, the vaccine is covered by medical insurance, with a majority of HCWs paying 10–20% of the price, which explains the much higher coverage rates compared to the low rates of vaccine acceptance if payment were needed.

Another strategy suggested to promote influenza vaccine uptake among HCWs relies on imposing enforced vaccination. In this study, slightly less than a half of the respondents (48%) agreed that vaccination of HCWs against influenza should be mandatory. This result points to the divisive nature of the mandatory vaccination issue. A discernible variability of attitude toward mandatory influenza vaccination among HCWs was displayed in a recent systematic review by Gualano et al., with positive attitudes reported in some settings (e.g., Saudi Arabia and Turkey) [110,111,112].

Conflicting data exist regarding enforcing influenza vaccination among HCWs. Specifically, a review by Wang et al. argued for the value of mandatory influenza vaccination in healthcare settings; nevertheless, the authors of the review demonstrated the prerequisites needed for effective implementation of this strategy [113]. These include the role of educational efforts, effective communication to focus on the value of such a strategy in patient safety and the significant role of support from the leadership in health institutions [114]. Additionally, in a review, Lytras et al. concluded that the most effective interventional measure to increase influenza vaccination coverage among HCWs is mandatory vaccination [115]. Moreover, Al Awaidy et al. found that a positive attitude toward mandatory influenza vaccination was correlated with higher rates of vaccine uptake among HCWs in Oman [46]. Furthermore, Helena Maltezou et al. recently argued for mandatory influenza vaccination among HCWs in light of the following reasons: (1) ease of implementation with lower costs, (2) sustainable accomplishment of high vaccination rates and (3) encouraging a culture of safety to prevail over autonomy [116].

On the contrary, De Serre et al. doubted the validity of compulsory influenza vaccination benefits based on the lack of hard evidence to show a clear positive impact of such a strategy on patient care [24]. Similarly, Edmond cited the lack of high-quality evidence with respect to mandatory influenza vaccination and subsequent threats against the employment status of HCWs; however, the author mentioned that the current evidence is sufficient to strongly recommend and encourage HCWs to accept and receive the influenza vaccine [117].

### Study Limitations

The results of the current study should be interpreted in light of several limitations as follows. First, the sampling approach with online recruitment of participants and chain referral could have resulted in a lack of randomization and potential selection bias; however, such an effect is likely minimal, considering the distribution of sex and occupational category in the study sample compared to the overall HCWs population in Jordan [55,56]. Second, the self-reported nature of influenza vaccine uptake assessment, with a lack of data from registry records on actual influenza vaccine coverage in Jordanian HCWs, is another caveat of the study. An important aspect that should be considered in any future work assessing vaccine hesitancy is that directionality of vaccine hesitancy/rejection and its associated psychological determinants cannot be inferred based a cross-sectional design. Therefore, hardwired vaccination intention might be a determinant of other tested variables, as observed in a recent study by Chambon et al. [118]. The lack of assessment of previous influenza vaccine uptake in influenza seasons prior 2021 is another caveat of the present study; however, our approach was adopted to minimize the possible effect of recall bias.

## 5. Conclusions

The findings of the current study can provide important clues to increase the influenza vaccine uptake among HCWs in Jordan. Consideration of psychological variables with respect to the intent to receive influenza vaccination is highly recommended, with communication of vaccine safety and emphasis on the potential risks of the disease. Promotion of patient safety culture to enhance the concept of collective responsibility is also encouraged. Reducing physical and psychological barriers, including cost issues, could be helpful in increasing influenza vaccine acceptance and uptake among HCWs in Jordan. As evidenced by the findings of this study, the role of educational programs appears highly valuable, considering the role of knowledge in influenza vaccine acceptance and uptake.

A multifaceted approach of interventional measures is needed to tackle the issue of influenza vaccine hesitancy among health professionals in Jordan. Although these measures might be challenging from a logistical point of view, the potential beneficial impact of improving vaccination rates against influenza can be used to promote such an approach. These measures include reducing constraints by offering vaccination on site and providing the vaccine for free. The issue of complacency can be tackled through educational campaigns highlighting the importance of vaccination in reducing HCW absenteeism and protecting vulnerable patients. Countering the embrace of general vaccine conspiracy beliefs can be highly valuable, considering its discernible role in vaccine uptake, as evidenced in this study.

## Figures and Tables

**Figure 1 vaccines-10-01355-f001:**
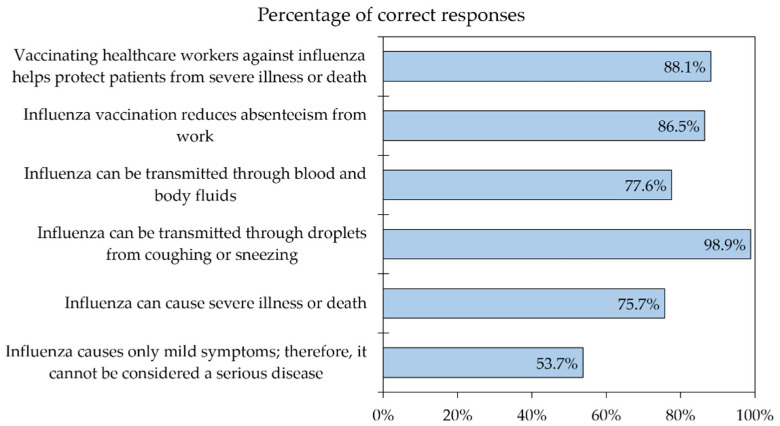
The overall level of knowledge about influenza among the study respondents (*n* = 1218).

**Figure 2 vaccines-10-01355-f002:**
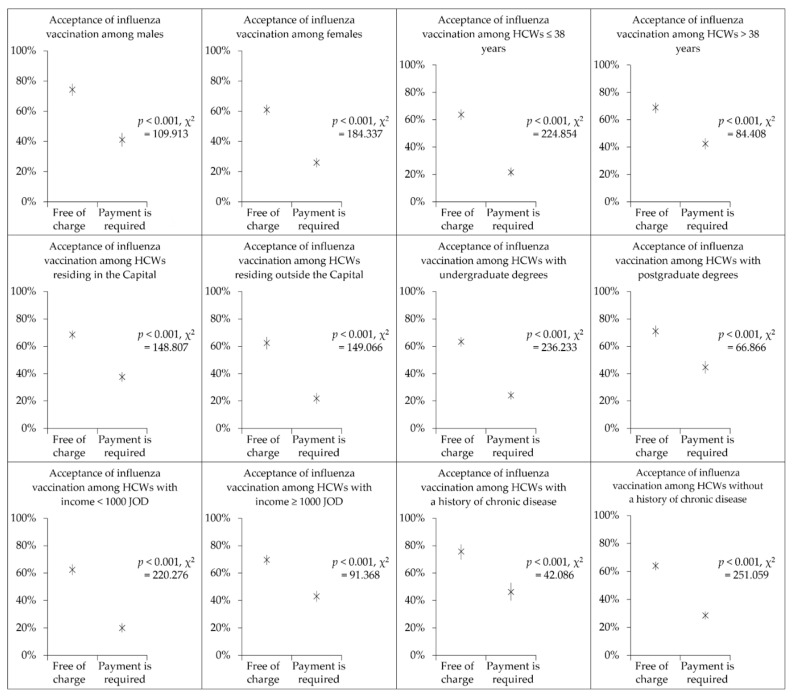
The decline in influenza vaccine acceptance if payment is required compared to acceptance of free vaccination stratified according to healthcare workers (HCW) variables. JOD: Jordanian dinar. ∗ symbols represent the mean estimate of vaccine acceptance per variable, whereas the lines represent the 95% confidence intervals of the estimates.

**Figure 3 vaccines-10-01355-f003:**
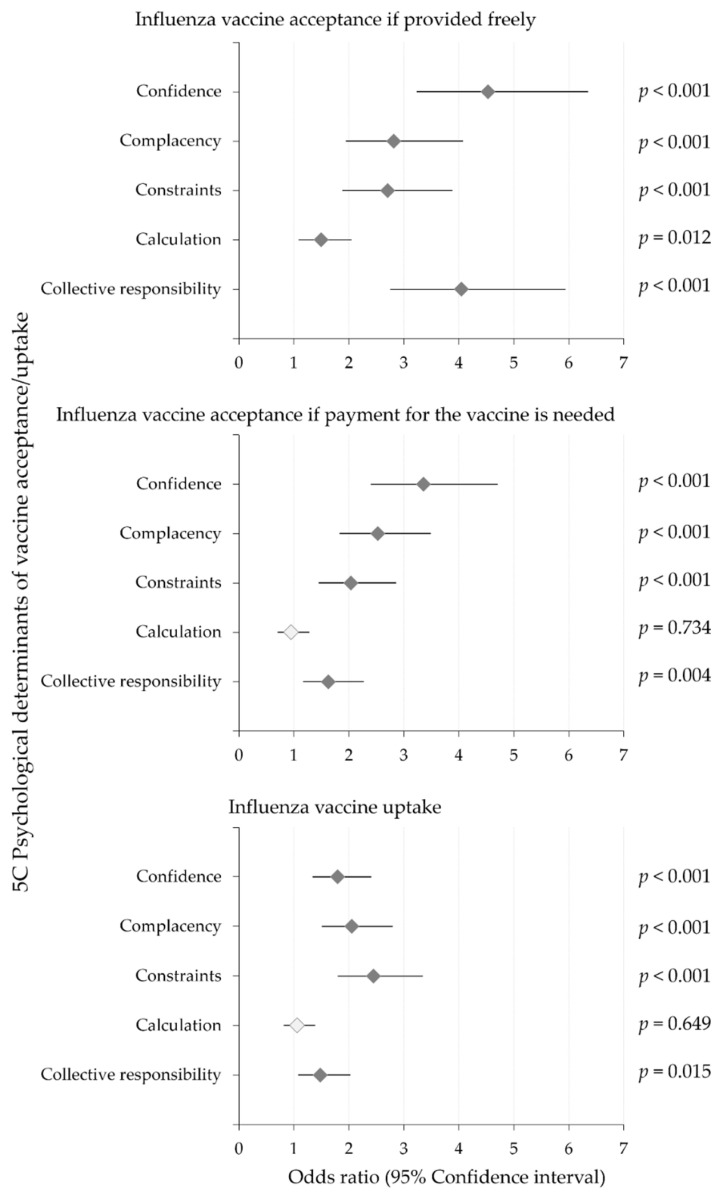
Multinomial regression analysis of the five psychological determinants (subscales) for influenza vaccine hesitancy and their association with influenza vaccine hesitancy (intent to receive influenza vaccination, with no/maybe responses). The mean odds ratios are represented by the gray diamond symbols, whereas the 95% confidence intervals are indicated by black bars.

**Table 1 vaccines-10-01355-t001:** Characteristics of the study sample divided by occupational category.

Characteristic			Occupational Category *n* ^3^ (%)					
		Total (*n =* 1218)	Nurse (*n* = 412, 33.8%)	Physician (*n* = 367, 30.1%)	Dentist (*n* = 87, 7.1%)	Pharmacist (*n* = 161, 13.2%)	Technician ^4^ (*n* = 182, 14.9%)	MSAR ^5^ (*n* = 9, 0.7%)
Median age (IQR) in years ^1^		38 (31–49)	36 (33–43)	49 (30–61)	45 (37–53)	37 (32–45)	34 (28–43)	37 (27–45)
Sex	Male	483 (39.7)	98 (23.8)	246 (67.0)	35 (40.2)	52 (32.3)	51 (28.0)	1 (11.1)
	Female	735 (60.3)	314 (76.2)	121 (33.0)	52 (59.8)	109 (67.7)	131 (72.0)	8 (88.9)
Region	Amman	777 (63.8)	188 (45.6)	296 (80.7)	55 (63.2)	111 (68.9)	122 (67.0)	5 (55.6)
	Outside Amman	441 (36.2)	224 (54.4)	71 (19.3)	32 (36.8)	50 (31.1)	60 (33.0)	4 (44.4)
Educational level	Undergraduate	754 (61.9)	351 (85.2)	128 (34.9)	50 (57.5)	102 (63.4)	115 (63.2)	8 (88.9)
	Postgraduate	464 (38.1)	61 (14.8)	239 (65.1)	37 (42.5)	59 (36.6)	67 (36.8)	1 (11.1)
Monthly income ^2^	JOD < 1000	590 (48.4)	309 (75.0)	70 (19.1)	22 (25.3)	76 (47.2)	105 (57.7)	8 (88.9)
	JOD ≥ 1000	628 (51.6)	103 (25.0)	297 (80.9)	65 (74.7)	85 (52.8)	77 (42.3)	1 (11.1)
History of chronic disease	Yes	231 (19.0)	53 (12.9)	118 (32.2)	15 (17.2)	25 (15.5)	18 (9.9)	2 (22.2)
	No	987 (81.0)	359 (87.1)	249 (67.8)	72 (82.8)	136 (84.5)	164 (90.1)	7 (77.8)

^1^ IQR: interquartile range; ^2^ JOD: Jordanian dinar; ^3^ *n*: number; ^4^ Technician: medical technicians, including professions in laboratory, radiology, rehabilitation and anesthesia; ^5^ MSAR: medical secretaries, administrators and receptionists.

**Table 2 vaccines-10-01355-t002:** The level of knowledge about influenza among the study respondents stratified by occupational category.

Influenza Knowledge/Attitude Item	Response	Occupational Category *n* ^2^ (%)						
		Nurse	Physician	Dentist	Pharmacist	Technician ^3^	MSAR ^4^	*p* Value ^5^
Influenza causes only mild symptoms; therefore, it cannot be considered a serious disease	Correct	242 (58.7)	165 (45.0)	47 (54.0)	92 (57.1)	103 (56.6)	5 (55.6)	0.005
	Incorrect	170 (41.3)	202 (55.0)	40 (46.0)	69 (42.9)	79 (43.4)	4 (44.4)	
Influenza can cause severe illness or death	Correct	269 (65.3)	330 (89.9)	67 (77.0)	116 (72.0)	132 (72.5)	8 (88.9)	<0.001
	Incorrect	143 (34.7)	37 (10.1)	20 (23.0)	45 (28.0)	50 (27.5)	1 (11.1)	
Influenza can be transmitted through droplets from coughing or sneezing	Correct	407 (98.8)	365 (99.5)	85 (97.7)	157 (97.5)	181 (99.5)	9 (100)	0.368
	Incorrect	5 (1.2)	2 (0.5)	2 (2.3)	4 (2.5)	1 (0.5)	0 (0)	
Influenza can be transmitted through blood and body fluids	Correct	344 (83.5)	295 (80.4)	67 (77.0)	106 (65.8)	126 (69.2)	7 (77.8)	<0.001
	Incorrect	68 (16.5)	72 (19.6)	20 (23.0)	55 (34.2)	56 (30.8)	2 (22.2)	
Influenza vaccination reduces absenteeism from work	Correct	343 (83.3)	350 (95.4)	77 (88.5)	139 (86.3)	139 (76.4)	5 (55.6)	<0.001
	Incorrect	69 (16.7)	17 (4.6)	10 (11.5)	22 (13.7)	43 (23.6)	4 (44.4)	
Vaccinating HCWs against flu helps protect patients from severe illness/death ^1^	Correct	344 (83.5)	352 (95.9)	76 (87.4)	146 (90.7)	148 (81.3)	7 (77.8)	<0.001
	Incorrect	68 (16.5)	15 (4.1)	11 (12.6)	15 (9.3)	34 (18.7)	2 (22.2)	
In Jordan, vaccination of HCWs against influenza should be mandatory	Yes	206 (50.0)	199 (54.2)	40 (46.0)	71 (44.1)	68 (37.4)	1 (11.1)	0.001
	No	206 (50.0)	168 (45.8)	47 (54.0)	90 (55.9)	114 (62.6)	8 (88.9)	

^1^ HCWs: healthcare workers, flu: influenza; ^2^ *n*: number; ^3^ Technician: medical technicians, including professions in laboratory, radiology, rehabilitation and anesthesia; ^4^ MSAR: medical secretaries, administrators and receptionists; ^5^ *p* value: calculated using the chi-squared test.

**Table 3 vaccines-10-01355-t003:** Characteristics of the study sample divided by attitude/practice with respect to influenza vaccination.

Characteristic		Would You Be Willing to Get the Influenza Vaccine if It Was Available for Free?			*p* Value, χ^2^	Are You Willing to Pay for Influenza Vaccine if the Cost Does Not Exceed JOD 15?			*p* Value, χ^2^	Have You Received the Influenza Vaccine in the Previous Year?		*p* Value, χ^2^
		Yes	Maybe	No		Yes	Maybe	No		Yes	No	
Sex	Male	359 (74.5)	84 (17.4)	39 (8.1)	<0.001, 25.930	198 (41.1)	140 (29.0)	144 (29.9)	<0.001, 37.273	330 (68.5)	152 (31.5)	0.001, 10.422
	Female	448 (61.6)	168 (23.1)	111 (15.3)		190 (26.1)	216 (29.7)	321 (44.2)		434 (59.7)	293 (40.3)	
Age	≤38 years	395 (64.4)	151 (24.6)	67 (10.9)	0.004, 10.889	134 (21.9)	201 (32.8)	278 (45.4)	<0.001, 61.387	355 (57.9)	258 (42.1)	<0.001, 16.047
	>38 years	412 (69.1)	101 (16.9)	83 (13.9)		254 (42.6)	155 (26.0)	187 (31.4)		409 (68.6)	187 (31.4)	
HCW category ^1^	Nurse	266 (64.6)	91 (22.1)	55 (13.3)	<0.001, 73.611	78 (18.9)	121 (29.4)	213 (51.7)	<0.001, 134.443	292 (70.9)	120 (29.1)	<0.001, 41.672
	Physician	300 (81.7)	43 (11.7)	24 (6.5)		188 (51.2)	97 (26.4)	82 (22.3)		249 (67.8)	118 (32.2)	
	Dentist	62 (71.3)	16 (18.4)	9 (10.3)		43 (49.4)	20 (23.0)	24 (27.6)		52 (59.8)	35 (40.2)	
	Pharmacist	90 (55.9)	44 (27.3)	27 (16.8)		40 (24.8)	54 (33.5)	67 (41.6)		85 (52.8)	76 (47.2)	
	Technician ^2^	89 (48.9)	58 (31.9)	35 (19.2)		39 (21.4)	64 (35.2)	79 (43.4)		86 (47.3)	96 (52.7)	
Region	Amman	532 (68.9)	144 (18.7)	96 (12.4)	0.036, 6.653	292 (37.8)	218 (28.2)	262 (33.9)	<0.001, 34.520	493 (63.9)	279 (36.1)	0.539, 0.378
	Out capital ^3^	275 (62.9)	108 (24.7)	54 (12.4)		96 (22.0)	138 (31.6)	203 (46.5)		271 (62.0)	166 (38.0)	
Educational level	Undergrad	477 (63.9)	178 (23.9)	91 (12.2)	0.003, 11.868	181 (24.3)	236 (31.6)	329 (44.1)	<0.001, 58.179	447 (59.9)	299 (40.1)	0.002, 9.745
	Postgrad	330 (71.3)	74 (16.0)	59 (12.7)		207 (44.7)	120 (25.9)	136 (29.4)		317 (68.5)	146 (31.5)	
Monthly income	JOD < 1000 ^4^	369 (63.4)	142 (24.4)	71 (12.2)	0.010, 9.297	118 (20.3)	178 (30.6)	286 (49.1)	<0.001, 85.862	357 (61.3)	225 (38.7)	0.136, 2.222
	JOD ≥ 1000	438 (69.9)	110 (17.5)	79 (12.6)		270 (43.1)	178 (28.4)	179 (28.5)		407 (64.9)	220 (35.1)	
Chronic disease	Yes	175 (76.4)	34 (14.8)	20 (8.7)	0.003, 11.512	107 (46.7)	64 (27.9)	58 (25.3)	<0.001, 31.219	171 (74.7)	58 (25.3)	<0.001, 16.565
	No	632 (64.5)	218 (22.2)	130 (13.3)		281 (28.7)	292 (29.8)	407 (41.5)		593 (60.5)	387 (39.5)	

^1^ HCW: healthcare worker (medical secretaries, administrators and receptionists were excluded due to their limited number); ^2^ Technician: medical technicians, including professions in laboratory, radiology and rehabilitation; ^3^ Out of capital: regions outside the capital, Amman; ^4^ JOD: Jordanian dinar.

## Data Availability

The data presented in this study are available on request from the corresponding author (Malik Sallam).

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
