# Peer review of "The Role of Psychological Factors and Vaccine Conspiracy Beliefs in Influenza Vaccine Hesitancy and Uptake among Jordanian Healthcare Workers during the COVID-19 Pandemic"

_vaccines, 2022, doi:10.3390/vaccines10081355_

Round 1
Reviewer 1 Report
The authors studied that the psychological factors and vaccine conspiracy beliefs in influenza vaccine hesitancy and uptake by a nation-wide survey in Jordanian during the COVID-19 pandemic. They targeted health care workers group in this study and recruited the participants using social media platforms, such as Facebook, Twitter, and Instagram. The survey as distributed in Arabic language. The survey items used in this study were basically adopted from previous similar studies including 5C psychological determinants of vaccination and VCBS (vaccine conspiracy beliefs scale).
They found that two-thirds of the study sample would accept influenza vaccination if provided freely. The strongest predictors of influenza vaccine acceptance were high levels of confidence and collective responsibility followed by low levels of complacency, constraints and calculation.
The results obtained from this study build on the authors' previous work and provide an advances by clarifying the factors needed to improve vaccination coverage, especially during the COVID-19 pandemic. This is considered a valuable paper for publication.
Author Response
Response: We are deeply thankful for the insightful summary and for the positive critical appraisal of the manuscript by the estimated reviewer. Thanks a lot.
Reviewer 2 Report
The study is adequate to respond to the objective of the study and has an adequate introduction, an approach to the problem that leads to generating an answer to the research question. I consider that the analysis is adequate, and the authors perform a sample size calculation and identify potential biases, which gives solidity to the results. Figure 1 does not provide additional information besides the percentage, which can be included together with the n of table 1. Same case for figure 2 and table 2; figure 3 and table 3. Figure 4 is not the best option, since the lines suggest that it is the same individuals who change their minds, rather than a decrease in the percentage of positive responses. I recommend removing the line and calculating the confidence intervals and including them in the plot (whiskers). In the analyses and figures corresponding to number 3.6, 3.7 and 3.8 sections, I could not identify the reference category for the OR in the text. It would be appropriate to make it explicit to improve understanding or to describe what would be the difference of interpreting it from that without a reference category. The discussion and conclusions I consider are appropriate to the results and objectives of the study. Finally, I recommend including a supplementary table in the core of the article if decide to eliminate some figures.
Author Response
Reviewer #2 Comments and Suggestions for Authors
- The study is adequate to respond to the objective of the study and has an adequate introduction, an approach to the problem that leads to generating an answer to the research question. I consider that the analysis is adequate, and the authors perform a sample size calculation and identify potential biases, which gives solidity to the results.
Response: We are grateful for the positive critical appraisal of the manuscript. Thanks a lot.
- Figure 1 does not provide additional information besides the percentage, which can be included together with the n of table 1.
Response: We agree with the reviewer regarding this point. Accordingly, we deleted (Figure 1) and added the percentages of HCWs categories to (Table 1).
Same case for figure 2 and table 2; figure 3 and table 3.
Response: Regarding Figure 2, it provided a graphical overview of the overall level of knowledge regarding influenza among the whole study sample; whereas in Table 2 we provided the results of comparison between different occupational categories. Therefore, we feel more inclined to keep both Figure 2 and Table 2 considering the differences in results they illustrate if the reviewer agrees. For Figure 3, and Table 3, we agree with the reviewer’s suggestion, and based on that we deleted Figure 3.
- Figure 4 is not the best option, since the lines suggest that it is the same individuals who change their minds, rather than a decrease in the percentage of positive responses. I recommend removing the line and calculating the confidence intervals and including them in the plot (whiskers).
Response: We would like to thank the reviewer for raising this important point and we agree with the suggestion. Accordingly, we revised the Figure based on the reviewer’s recommendation.
- In the analyses and figures corresponding to number 3.6, 3.7 and 3.8 sections, I could not identify the reference category for the OR in the text. It would be appropriate to make it explicit to improve understanding or to describe what would be the difference of interpreting it from that without a reference category.
Response: We agree with the reviewer’s suggestion, and to clarify this point, we made the revisions to include the reference categories.
- The discussion and conclusions I consider are appropriate to the results and objectives of the study. Finally, I recommend including a supplementary table in the core of the article if decide to eliminate some figures.
Response: We are deeply thankful for the positive review of the manuscript. Regarding the suggested supplementary Table, we feel that the tables provided currently provide the details needed as suggested in the point #2 & #3. Please check the revised highlighted manuscript.
Reviewer 3 Report
Thank you for the opportunity to read the paper "The role of psychological factors and vaccine conspiracy beliefs in influenza vaccine hesitancy and uptake among Jordanian healthcare workers during the COVID-19 pandemic" submitted to Vaccines. The paper addresses an important issue - factors contributing to the intention to be vaccinated and perceived barriers to vaccination in a country that is not yet been represented adequately in the published literature (Jordan).
This is a high quality manuscript. The manuscript is well written, the sample size is adequate, the analyses are state of the art. The paper has the potential to make a valuable contribution to the field. There are, however, a couple of points that I would like to see addressed before I can recommend acceptance.
1. The authors write that the link to participate in the study was distributed via WhatsApp (among other channels). Does that mean that anyone who had access to the link could participate? How was it ensured that really only health care workers from Jordan participated in the survey?
2. Although the authors present a sample size calculation, they do not reveal how they do not explain how the arrived at the estimations on which the calculation is based. This is the most important part of the power analysis/sample size calculation. Please provide this explanation in the revised manuscript.
3. On page 3, the systematic review by Schmid et al. is discussed in some detail. The relevance for the study could be elaborated more strongly instead of simply summarising the findings.
4. It has been shown that conspiracy beliefs are a decisive factor why people do not get vaccinated. The authors acknowledge this, but discuss mainly work from their own research group in this context. The manuscript would benefit from a more balanced presentation at this point. See for example these highly relevant present articles:
Imhoff, R., & Lamberty, P. (2020). A bioweapon or a hoax? The link between distinct conspiracy beliefs about the coronavirus disease (COVID-19) outbreak and pandemic behavior. Social Psychological and Personality Science, 11(8), 1110–1118.
Ripp, T., Röer, J.P. (2022). Systematic review on the association of COVID-19-related conspiracy belief with infection-preventive behavior and vaccination willingness. BMC Psychology, 10, 66.
Salali, G. D., & Uysal, M. S. (2020). COVID-19 vaccine hesitancy is associated with beliefs on the origin of the novel coronavirus in the UK and Turkey. Psychological Medicine, 1-3.
5. The authors write that participants with "apparently careless responses" were excluded from the analyses. In the supplementary figure, it can be seen that 33 participants were excluded, because they did not answer with N/A to the question "Do you recommend influenza vaccination for the patients?". Why were these participants excluded? This is not clear to me.
6. It is very unusual for authors to be referred to by their first and last names in the text. This should be changed.
7. Figure 2 shows that only just over half of the participants knew the correct answer to question 6. This seems to me to be a very poor result. Why is that? Do the authors have an explanation for this?
Author Response
Reviewer #3 Comments and Suggestions for Authors
- Thank you for the opportunity to read the paper "The role of psychological factors and vaccine conspiracy beliefs in influenza vaccine hesitancy and uptake among Jordanian healthcare workers during the COVID-19 pandemic" submitted to Vaccines. The paper addresses an important issue - factors contributing to the intention to be vaccinated and perceived barriers to vaccination in a country that is not yet been represented adequately in the published literature (Jordan). This is a high quality manuscript. The manuscript is well written, the sample size is adequate, the analyses are state of the art. The paper has the potential to make a valuable contribution to the field. There are, however, a couple of points that I would like to see addressed before I can recommend acceptance.
Response: We are deeply grateful for the positive critical appraisal of the manuscript and the nice summary. Thanks a lot.
- The authors write that the link to participate in the study was distributed via WhatsApp (among other channels). Does that mean that anyone who had access to the link could participate? How was it ensured that really only health care workers from Jordan participated in the survey?
Response: We are thankful for this important point; however, we believe that the introductory section of the survey was explicit regarding the target population of the study; namely, HCWs in Jordan. In addition, the sampling approach was based on the circulation of the survey among the HCWs in the country since the start of the sampling was based on the contacts of the authors in Jordan.
Since we believe that the point raised by the reviewer is relevant and important, we added the following statement to the Methods section: “The inclusion criteria as explicitly mentioned in the introductory section of the survey link were: (1) Jordanian HCWs…”
- Although the authors present a sample size calculation, they do not reveal how they do not explain how the arrived at the estimations on which the calculation is based. This is the most important part of the power analysis/sample size calculation. Please provide this explanation in the revised manuscript?
Response: We would like to thank the reviewer for highlighting this point. The proportion was assumed at the level of 0.5 considering the lack of recent estimates on the influenza vaccine acceptance among Jordanian HCWs, and a precision level of 0.03 was decided to reduce the margin of error in our estimates. Based on the reviewer’s important point, we added the following reference that justifies our approach for the assumptions made for the estimation of the minimum sample size: 58. Sapra, R.L. How to Calculate an Adequate Sample Size? In How to Practice Academic Medicine and Publish from Developing Countries? A Practical Guide, Nundy, S., Kakar, A., Bhutta, Z.A., Eds. Springer Nature Singapore: Singapore, 2022; 10.1007/978-981-16-5248-6_9pp. 81-93.
- On page 3, the systematic review by Schmid et al. is discussed in some detail. The relevance for the study could be elaborated more strongly instead of simply summarising the findings.
Response: We totally agree with the reviewer’s point regarding the relevance and importance of the comprehensive review by Schmid et al. Accordingly, we added the following paragraph to the Introduction section: “The relevance of this review is related to the thorough presentation of possible barriers to influenza vaccination on a micro and macro levels [43]. Subsequently, the review establishes of a clear direction for future research that can help in the design of evidence-based intervention measures to address influenza vaccination hesitancy [43].”
- It has been shown that conspiracy beliefs are a decisive factor why people do not get vaccinated. The authors acknowledge this, but discuss mainly work from their own research group in this context. The manuscript would benefit from a more balanced presentation at this point. See for example these highly relevant present articles:
Imhoff, R., & Lamberty, P. (2020). A bioweapon or a hoax? The link between distinct conspiracy beliefs about the coronavirus disease (COVID-19) outbreak and pandemic behavior. Social Psychological and Personality Science, 11(8), 1110–1118.
Ripp, T., Röer, J.P. (2022). Systematic review on the association of COVID-19-related conspiracy belief with infection-preventive behavior and vaccination willingness. BMC Psychology, 10, 66.
Salali, G. D., & Uysal, M. S. (2020). COVID-19 vaccine hesitancy is associated with beliefs on the origin of the novel coronavirus in the UK and Turkey. Psychological Medicine, 1-3.
Response: We would like to thank the reviewer for this important suggestion. We totally agree that the references listed would be highly valuable in the discussion regarding the possible association of conspiracy beliefs with intention to get vaccinated and the actual uptake of the vaccine, particularly in relation to the rapid rate of emergence of infectious diseases (e.g. Ebola, COVID-19 and currently the ongoing human monkeypox outbreak), and the endorsement of conspiracy beliefs can have a negative impact on health-seeking behaviour including willingness to receive the vaccines. Based on the important suggestion by the reviewer, we added the following paragraph citing the references suggested: “The frequent emergence of infectious diseases accompanied by the wide prevalence of adopting conspiratorial narratives has become a notable observation (e.g. during the COVID-19 pandemic and the ongoing multi-country monkeypox outbreak) [104-107]. The negative impact of conspiracy beliefs that were related to COVID-19 pandemic has been presented comprehensively in a recent systematic review by Ripp and Röer [104]. This included the reduced willingness to get vaccinated besides the negative association with the preventive behavior (e.g. adhering to social distancing guidelines) [104,108]. In this study, the negative impact of endorsing vaccine conspiracy beliefs was manifested in the association with lower rates of influenza vaccine uptake in the study sample highlighting the need to consider the aspects of conspiracies in the implementation of effective intervention measures aiming to promote vaccination [104,109].”
- Ripp, T.; Röer, J.P. Systematic review on the association of COVID-19-related conspiracy belief with infection-preventive behavior and vaccination willingness. BMC Psychology 2022, 10,(1): 66, doi:10.1186/s40359-022-00771-2.
- Imhoff, R.; Lamberty, P. A Bioweapon or a Hoax? The Link Between Distinct Conspiracy Beliefs About the Coronavirus Disease (COVID-19) Outbreak and Pandemic Behavior. Social Psychological and Personality Science 2020, 11,(8): 1110-1118, doi:10.1177/1948550620934692.
- Salali, G.D.; Uysal, M.S. COVID-19 vaccine hesitancy is associated with beliefs on the origin of the novel coronavirus in the UK and Turkey. Psychological Medicine 2020, Online ahead of print,(n/a): 1-3, doi:10.1017/S0033291720004067.
- Sallam, M.; Al-Mahzoum, K.; Dardas, L.A.; Al-Tammemi, A.a.B.; Al-Majali, L.; Al-Naimat, H.; Jardaneh, L.; AlHadidi, F.; Al-Salahat, K.; Al-Ajlouni, E., et al. Knowledge of Human Monkeypox and Its Relation to Conspiracy Beliefs among Students in Jordanian Health Schools: Filling the Knowledge Gap on Emerging Zoonotic Viruses. Medicina 2022, 58,(7): 924, doi:10.3390/medicina58070924.
- Bierwiaczonek, K.; Kunst, J.R.; Pich, O. Belief in COVID-19 Conspiracy Theories Reduces Social Distancing over Time. Appl Psychol Health Well Being 2020, 12,(4): 1270-1285, doi:10.1111/aphw.12223.
- Knight, H.; Jia, R.; Ayling, K.; Bradbury, K.; Baker, K.; Chalder, T.; Morling, J.R.; Durrant, L.; Avery, T.; Ball, J.K., et al. Understanding and addressing vaccine hesitancy in the context of COVID-19: development of a digital intervention. Public Health 2021, 201,(n/a): 98-107, doi:10.1016/j.puhe.2021.10.006.
- The authors write that participants with "apparently careless responses" were excluded from the analyses. In the supplementary figure, it can be seen that 33 participants were excluded, because they did not answer with N/A to the question "Do you recommend influenza vaccination for the patients?". Why were these participants excluded? This is not clear to me.
Response: We would like to thank the reviewer for the opportunity to clarify this issue. One of the criteria used to exclude the participants for possible careless responses was based on the fact that medical secretaries, administrators and receptionists (MSAR) are not responsible for recommending vaccination based on their role that does not entail a direct role in patient care and advice. Thus, we excluded those who did not provide the response (not applicable) for this item as indicated in the supplementary figure.
- It is very unusual for authors to be referred to by their first and last names in the text. This should be changed.
Response: Thanks for this comment and accordingly we revised the in-text reference to the authors referring only to the senior names of the authors. Please check the revised highlighted manuscript.
- Figure 2 shows that only just over half of the participants knew the correct answer to question 6. This seems to me to be a very poor result. Why is that? Do the authors have an explanation for this?
Response: We would like to thank the reviewer for pointing to this result. However, we have already mentioned this point in the discussion in the context of complacency as a psychological determinant of vaccine acceptance/uptake. Please check the revised highlighted manuscript (lines 535-539): “The impact of complacency in this study can be related to the finding of a slightly less than half of the study group stating that influenza causes mild symptoms only; therefore, it cannot be considered as a serious disease. Thus, improving the level of disease knowledge can increase influenza vaccine acceptance and enhance its uptake among HCWs.”